# Competitive plasticity to reduce the energetic costs of learning

**Mark C. W. van Rossum**[1,2]*, **Aaron Pache**[2]

**1** School of Psychology, University of Nottingham, Nottingham, United Kingdom, **2** School of Mathematical Sciences, University of Nottingham, Nottingham, United Kingdom

* mark.vanrossum@nottingham.ac.uk

## Abstract

The brain is not only constrained by energy needed to fuel computation, but it is also constrained by energy needed to form memories. Experiments have shown that learning simple conditioning tasks which might require only a few synaptic updates, already carries a significant metabolic cost. Yet, learning a task like MNIST to 95% accuracy appears to require at least $10^8$ synaptic updates. Therefore the brain has likely evolved to be able to learn using as little energy as possible. We explored the energy required for learning in feedforward neural networks. Based on a parsimonious energy model, we propose two plasticity restricting algorithms that save energy: 1) only modify synapses with large updates, and 2) restrict plasticity to subsets of synapses that form a path through the network. In biology networks are often much larger than the task requires, yet vanilla backprop prescribes to update all synapses. In particular in this case, large savings can be achieved while only incurring a slightly worse learning time. Thus competitively restricting plasticity helps to save metabolic energy associated to synaptic plasticity. The results might lead to a better understanding of biological plasticity and a better match between artificial and biological learning. Moreover, the algorithms might benefit hardware because also electronic memory storage is energetically costly.

**Data Availability Statement:** Code is present at github.com/vanrossumlab/frugal_24.

**Funding:** The author(s) received no specific funding for this work.

## Author summary

There is increasing evidence learning already very simple tasks in animals requires substantial amounts of metabolic energy. This raises the question how much energy it costs to learn more complex tasks. For instance, the well known backpropagation algorithm always updates all synapses in the network and one can wonder if that is necessary. In this theoretical study we estimate the energy needed for an artificial neural network to learn to classify the well-known MNIST data set. We find that in particular in larger networks, substantial energy savings can be achieved by carefully selecting which synapses to update. This is particular relevant for the neural networks in the brain that often contain millions of neurons. This study will hopefully lead to a better understanding of learning processes in biology. Moreover, as training large artificial neural networks consumes substantial

**Competing interests:** The authors have declared that no competing interests exist.

amount of electric energy, the savings algorithms proposed here, might help to reduce those costs as well.

## Introduction

Energy availability is a vital necessity for biological organisms. The nervous system is a particularly intensive energy consumer. Though it constitutes approximately 2% of human body mass, it is responsible for roughly 20% of basal metabolism—continuously consuming some 20W [1, 2]. Energy requirements of neural signaling are now widely seen as an important design constraint of the brain. Sparse codes and sparse connectivity (pruning) can be used to lower energy requirements [3]. Learning rules have been designed that yield energy efficient networks [4, 5].

However, learning itself is also energetically costly: in classical conditioning experiments with flies the formation of long-term memory reduced lifespan by 20% when the flies were subsequently starved [6]. Moreover, starving fruit flies halt long-term memory formation; nevertheless forcing memory expression reduced their lifespan [7]. We have estimated the cost of learning in fruitflies at 10mJ/bit [8]. These costs are not restricted to flies. In mammals there is physiological evidence that energy availability gates long lasting forms of plasticity [9]. Furthermore, in humans there is behavioral evidence for a correlation between metabolism and memory [10, 11].

Given this evidence, we hypothesize that biological neural plasticity is designed to be energy efficient. Including energy constraints could lead to computational learning rules more closely resembling biology, and a better understanding of observed synaptic plasticity rules. As precise metabolic cost models for plasticity are not yet available, we aim at algorithmic principles rather than energy efficient biophysical mechanisms. We introduce an abstract energy measure that assumes the energy is solely used by the synaptic updates. Costs for synaptic transmission and global costs independent of the number of synapses are ignored (see Discussion). We examine how this energy measure can be reduced while maintaining overall learning performance.

Our study is inspired by experimental observations that despite common fluctuations in synaptic strength, the number of synapses undergoing permanent modification appears restricted. First plasticity is spatially limited, e.g. during motor learning plasticity is restricted to certain dendritic branches [12]. Second, plasticity is temporally limited, e.g. it is not possible to induce late-phase LTP twice in rapid succession [13, 14]. This stands in stark contrast with traditional backprop which updates *every* synapse on *every* trial and, as we shall see, can lead to very inefficient learning.

We use artificial neural networks as a model of neural learning. While neural networks trained with back-propagation are an abstraction of biological learning, it allows for an effective way to teach networks complex associations that is currently not matched by more biological networks or algorithms. Biological implementations of back-propagation have been suggested, e.g. [15], but it is likely that these are less energy efficient as learning times are typically longer.

In artificial neural network research there is a similar interest in limiting plasticity, but typically for other reasons. Carefully selecting which synapses to modify can prevent overwriting previously stored information, also known as catastrophic forgetting [16]. Randomly switching plasticity on and off can help regularization, as in drop-out and its variants [17]. Communication with the memory systems is also energetically costly in computer hardware. Strikingly,

storing two 32-bit numbers in DRAM is far more expensive than multiplying them [18]. Recent studies have started to explore algorithms that reduce these costs by limiting the updates to weight matrices [18, 19].

## Methods

### Data-set

As data set we use the standard MNIST digit classification task. In the MNIST task 28x28 grey-scale images of handwritten digits need to be classified according to the digit they represent ('0', '1', '2', . . .). Similar results were found on the fashionMNIST data set. The data were offset so that the mean input was zero. This common pre-processing step is important. Namely, when trained with backprop, the weight update to weight $w_{ij}$ between input $x_j$ to a unit $i$ with error $\delta_i$ is

$$\Delta w_{ij} = -\epsilon \delta_i x_j$$

where $\epsilon$ is the learning rate which was adjusted to approximately minimize energy consumption (see below). For a mean square error cost function, the error in the output layer is $\delta_i = g'(h_i)(y_i - t_i)$, where $h_i$ is the net input to the unit, $g()$ the activation function, $y$ the unit's output, and $t$ it's target value; while in the hidden layer one uses the back-propagated error $\delta_i = g'(h_i) \Sigma_j w_{ij} \delta_j$.

As the weight update is proportional to the input value $x_j$, there is no plasticity for zero-valued inputs, even when $\delta_i \neq 0$. However, this could be a confounding factor as we would like to be in full control of the number of plasticity events. After zero-meaning the inputs, only a negligible number of inputs will be exactly zero and this issue does not arise. Reassuringly, the relative savings achievable are not substantially affected by this assumption, Fig A in S1 Text.

### Network architecture

We used a network with 784 input units (equal to the number of MNIST pixels) with bias, a variable number of hidden units in a single hidden unit layer, and 10 output units (one for each digit), Fig 1a. Networks had all-to-all, non-convolutional connectivity. The hidden layer units used a leaky rectifying linear activation function (lReLU), so that $g(x) = x$ if $x \geq 0$, and otherwise $g(x) = \beta x$ with $\beta = 0.1$. A common alternative activation function is the rectified linear function, $g(x) = \max(0, x)$. However, this activation function would lead to a substantial fraction of neurons with zero activation in the hidden layer on a given sample and turn off plasticity of many synapses between hidden and output layer. As above, this would potentially be confounding, but it is avoided by using 'leaky' units. This did not substantially change the savings achievable, Fig A in S1 Text.

### Training

The activation of the ten output units is used to train the network. The target distribution was one-hot encoding of the image labels. For classification tasks it is common to apply a soft-max non-linearity $y_i = \exp(h_i)/\Sigma_j \exp h_j$, where $h_i$ is the net input to each unit, so that the output activities represent a normalized probability distribution. One then trains the network by minimizing the cross entropy loss between the output distribution and the target distribution.

However, it is easy to see that when the target is a deterministic one-hot (0 and 1) distribution, the cross entropy loss only vanishes in the limit of diverging $h_i$, which in turn requires diverging weights. As cross-entropy loss minimization is therefore energy inefficient, we used

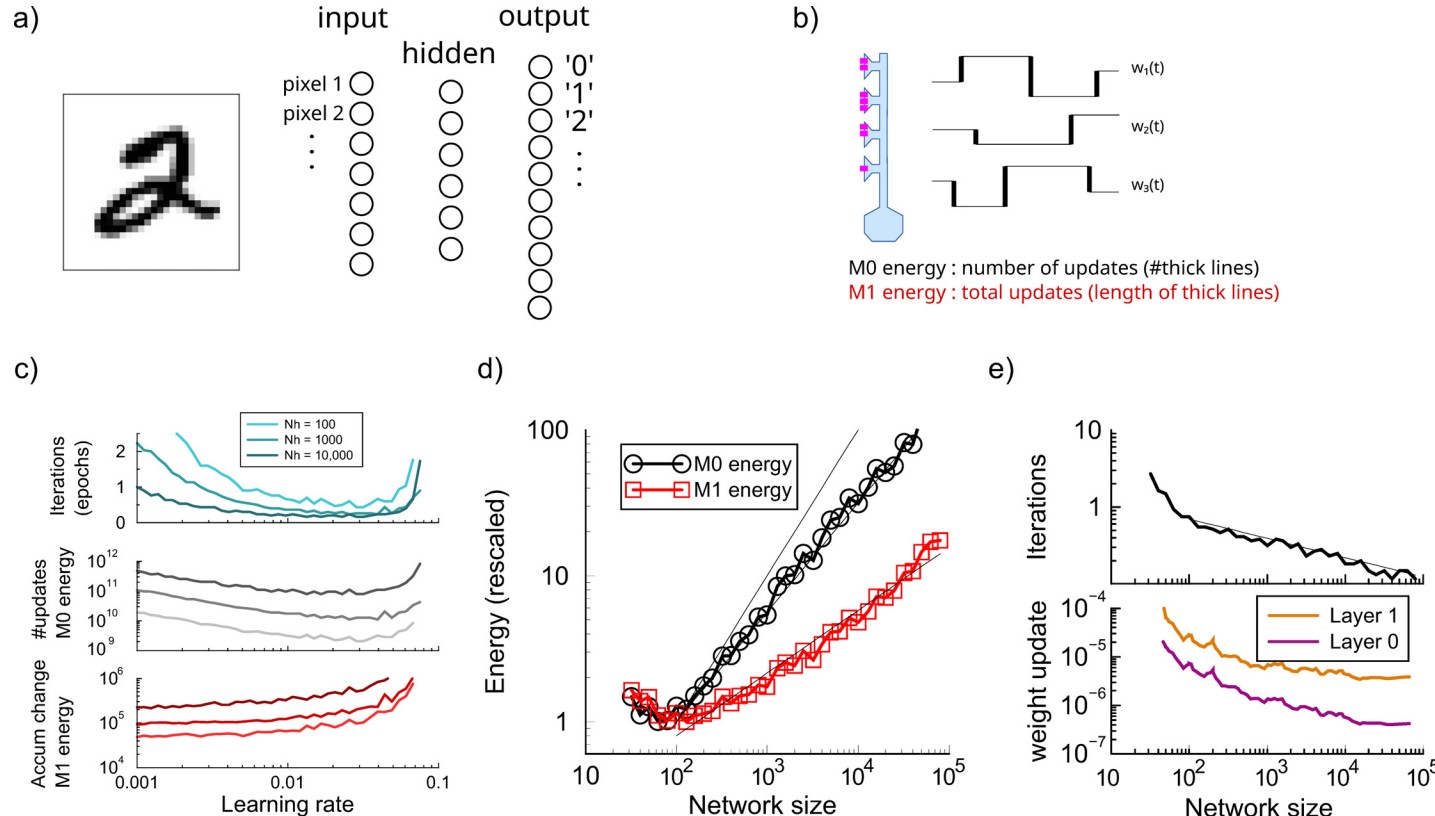

**Fig 1. Energy requirements to train large standard backpropagation networks.** a) Network setup has a single hidden layer with a variable number of hidden units. b) Definition of the $M_0$ and $M_1$ of energy measures. c). Energy required to train a network with 100, 1000, or 10000 hidden units as a function of the learning rate to reach 95% accuracy. Top panel: number of iterations required. Middle panel: The total number of updates ($M_0$ energy) is minimal when the number of iterations is minimal. Bottom panel: The accumulated changes ($M_1$ energy) is lowest in the limit of small learning rates. d). Both energy measures versus network size. Since the network has one hidden layer, network size is expressed in the number of hidden units. Black curve is the number of updates ($M_0$), red is the sum of all absolute changes ($M_1$); both grow in large networks. The y-axis was scaled so that the minimum energy was one. The three thin lines represent slope one and powerlaw fits (see text). e). Learning speed and update size vs network size. Top: Large networks train somewhat faster to the 95% accuracy criterion than small networks. Bottom: the mean absolute weight update is smaller in large networks.

linear output units and minimized the mean squared error between output and target. This did not substantially change the achievable savings, Fig A in S1 Text.

We used the backpropagation learning algorithm to train the network. Weights were initialized with small Gaussian values ($\sigma = 0.01$). The energy measures depend only weakly on it, but very small values will slow down initial training and very large values would require learning to reduce them. In preliminary simulations, regularization was seen to have no significant effect on energy or savings, and was subsequently omitted. Samples were presented in a random order. Because batching would require storing synaptic updates across samples, which would be biologically challenging, the training was not batched and the synaptic weights were updated after every sample. We did not find substantial energy savings when using adaptive learning schemes such as ADAM. As such schemes would furthermore be hard to implement biologically, we instead fixed the learning rate.

Every 1000 samples training was interrupted and performance on a validation data set was measured. As a reasonable compromise between performance and compute requirements, networks were always trained until a validation set accuracy criterion of 95% was achieved.

### Estimating the energy needed for plasticity

Our approach requires a model of the metabolic energy needed for learning. It is currently not known why the metabolic cost of some forms of plasticity is so high, nor is it known which process is the main consumer. In flies persistent, protein synthesis dependent memory is much more costly than protein synthesis independent forms of memory. One could presume that protein synthesis itself is costly, however it has also been argued that protein synthesis is relatively cheap [20]. After all, also in the absence of synaptic plasticity there is substantial protein turnover. Examples of other costly processes could be transport of proteins to synapses, actin thread-milling, synaptogenesis. In addition, there might be energy costs that are not related to synaptic plasticity, such as replay (see Discussion); we ignore those here. Furthermore, more biophysical energy models would require spiking networks of excitatory and inhibitory neurons, which is out of the scope of the current study.

Instead, we propose a generic model for the metabolic energy $M$ of synaptic plasticity under the following assumptions:

1) While it is well known that there are interactions between plasticity of nearby synapses, e.g. [21, 22], we assume that there is no spatial interaction in regards to synaptic costs. It might be that potentiation of two synapses is relatively cheaper than potentiation of a single synapse, e.g. when costly pathways can be shared, as in synaptic tagging and capture [23–25]. However, instead it is also possible that there is competition for resources [26–28], and potentiation of two nearby synapses could be extra costly [29]. The interactions when one synapse undergoes potentiation while another one undergoes depression, might even be more complicated as resources might be reused. In summary, both cooperative as well as competitive interactions likely exist but as there is currently no information about their energetics, we need to ignore them. This implies that the energy is the sum of the individual costs, $M = \sum_{i,j} M^{(i,j)}$, where $M^{(i,j)}$ is the energy required to modify synapse $(i, j)$.

2) Similar arguments can be made about the temporal interactions between plasticity. Again, there is extensive literature on temporal interactions in plasticity induction and expression [14, 30, 31]. Energetically, it might be cheaper to potentiate a synapse twice in rapid succession, but it could equally be more costly. We assume that there is no temporal interaction in regards to synaptic costs. This implies that the total metabolic cost is summed over all timesteps $M = \sum_t M(t)$.

3) Next, we assume that synaptic potentiation and depression both incur the same cost. As we are typically interested in savings in energy between algorithms, this assumption turns out to be minor. Consider a case where, say, synaptic potentiation costs energy but depression does not. As synapses undergo a similar number of potentiation and depression events during training, such a variant would halve the cost, however, it does not substantially change the achievable savings, Fig A in S1 Text. Instead, it is the endless up and down of the synaptic weights that makes learning inefficient.

Under the assumption of spatial and temporal independence, we still require an expression how much modification of a single synapse costs. We propose that this scales as $|\Delta w|^{\alpha}$, where $\alpha$ is a parameter. The total energy sums across all neurons and all training time steps

$$M_{\alpha} = \sum_{i,j,t} \left| \Delta w_{i,j}(t) \right|^{\alpha}$$

The parameter $\alpha$ expresses the proportionality in the weight change. When $\alpha = 1$, the energy ($M_1$) is linear in the amount of weight change and represents the accumulated weight changes,

see Fig 1a for a graphical representation. This is for instance relevant for the energy consumed by protein synthesis where larger changes would require more proteins. Values of $\alpha > 1$ would lead to a situation where updating a synapse twice would be cheaper than updating once with double the amount. This is not impossible, but seems unlikely.

In the limit $\alpha \to 0$, $M_\alpha$ counts the total number of synaptic updates, irrespective of their size, Fig 1a. As an example, this would be the case when synaptic tag setting would be costly [24]. It is also a reasonable cost function for digital computer architectures, where the cost of writing a memory is independent of its value. While intermediate values of $\alpha$ ($0 < \alpha < 1$) as well as combinations of terms are possible, we concentrate on $M_0$ and $M_1$.

The energy measures are summed across both layers of the network. The number of synapses of both layers is proportional to the number of hidden units. But as dictated by the task, the input-to-hidden layer has 784 synapses per hidden unit, while the hidden-to-output layer has only 10 synapses per hidden unit. In unmodified networks the plasticity cost of the input-to-hidden layer therefore dominates the total energy.

## Results

### Energy requirements of learning

We first explored the energy requirements of training standard large networks. We compared the training of a network with 100, 1000, and 10000 hidden units, Fig 1c. In addition to the number of iterations needed to reach criterion performance (top panel), we measured the two energy variants described in the Methods. The total number of synaptic updates, called $M_0$, is directly proportional to the learning time. If the learning rate is too low, learning takes too long; too high and learning fails to converge. An intermediate learning rate minimizes the learning time and hence $M_0$ energy, Fig 1c; middle panel. The optimal learning rate is in good approximation independent of network size.

The cumulative weight change, called $M_1$ energy, measures the total amount of absolute weight changes of all synapses across training, Fig 1c; bottom panel. This energy measure is smallest in the limit of small learning rates, where it becomes independent of the learning rate. With a small learning rate the path in weight space is more cautious without the overshooting associated to larger learning rates. While the energy measures $M_0$ and $M_1$ thus have a different optimal learning rate, below we use a learning rate of 0.01 as a compromise, allowing direct comparison.

### Cost of learning in large networks

The MNIST classification networks typically have layers consisting of some hundred units. However, as in biology the number of neurons is enormous (see Discussion), we examine the energy consumed as a function of the network size, Fig 1d. First consider the total number of updates, $M_0$. Because our setup rules out that synaptic updates are accidentally exactly zero (see Methods), the number of non-zero updates is the total number of synapses in the network multiplied by the number of iterations $T$. Larger networks learn a bit quicker than smaller ones, approximately scaling as $T \propto N_h^{-0.249}$, Fig 1e top, where $N_h$ denotes the number of units in the hidden layer. As a result, energy scales sub-linearly with network size ($M_0 \propto N_h T \propto N_h^{0.75}$).

The accumulated synaptic change energy $M_1$ also increases for larger networks albeit less steeply, Fig 1d. Despite a fixed learning rate, the size of individual weight updates is smaller in larger networks, Fig 1c, bottom, see also [32]. In summary, in large networks individual

synapses 1) take fewer and smaller steps, and 2) the final individual weights are smaller. We find an approximate square root relation between energy and size, $M_1 \propto N_h^{0.43}$, Fig 1d.

## Randomly restricting plasticity to save energy

The above result shows that while large, over-dimensioned networks learn a bit faster, training them uses far more energy—restricting plasticity might save energy. To examine this we first only allow plasticity in a random subset of synapses using a random binary mask. Mask elements were drawn from a Bernoulli distribution. We first used a different random mask for every trial, also known as 'Dropconnect' [17]. In this case learning simply slowed down and masking did not save energy; also using a different mask specific for each output class did not save energy.

Next, the mask was fixed throughout training. Now both energy measures strongly reduced, Fig 2a, green curves. The optimal mask density was estimated from a counting argument. (We also numerically optimized the fraction of plastic synapses in the input layer; this gave similar results). It is based on the above observations that to achieve the criterion performance one needs at least about 100 hidden units in case that all connections are plastic, Fig 1d. In other words, there need to be about $\mu n_{in} n_{out}$ plastic paths between any input and any output, where $n_{in} = 28^2$ is the number of inputs and $n_{out} = 10$ the number of output units, and $\mu = 100$. We now assume that this is the critical property of the network. According to the hypergeometric distribution, random masks in the input and output layer reduce the mean number of plastic paths to $\mu = f_0 f_1 N_h$, where $f_i$ denotes the fraction of plastic connections in layer $i$.

At the optimal probability, we find that the weight change is approximately the same across network size and layer. Hence both the $M_0$ and $M_1$ energy are proportional to $m(f_0, f_1) = n_{in} N_h f_0 + n_{out} N_h f_1$, We minimize $m$, subject to $\mu = f_0 f_1 N_h$ and $0 \leq f_i \leq 1$. For a given $\mu \geq N_h$, the energy is minimal when $f_0^* = \frac{\mu}{N_h} \max\left(1, \sqrt{\frac{N_h n_{out}}{\mu n_{in}}}\right)$ and $f_1^* = \min\left(1, \sqrt{\frac{\mu n_{in}}{N_h n_{out}}}\right)$, see Fig 2b.

The interpretation is that because the number of inputs by far exceeds the number of outputs, it is optimal to keep all hidden-to-output connections plastic ($f_1 = 1$). Only in very large networks when $N_h \gtrsim \mu n_{in}^2 / n_{out} \approx 8000$, it becomes wasteful to keep all outgoing connections plastic and it is best to reduce the fraction of plastic output synapses as well. The energy scales as

$$m(f_0^*, f_1^*) = \mu n_{in} \max\left(1, \sqrt{\frac{N_h n_{out}}{\mu n_{in}}}\right) + N_h n_{out} \min\left(1, \sqrt{\frac{\mu n_{in}}{N_h n_{out}}}\right) \tag{1}$$

The energy according to this estimate is plotted as the dashed curve in Fig 2, with the proportionality constant extracted from the value of the energy at $N_h = 100$. It gives a decent fit. The energy increases in large networks as not every plastic connection in the input-to-hidden layer is met with a plastic connection in the hidden-to-output layer. For large networks both energy measures scale as $m \propto \sqrt{N_h}$.

A similar approach is to mask on a per neuron basis and only allow plasticity in a fixed subset of neurons in the hidden layer, Fig 2. Hereto we multiply the vector of back-propagated errors element-wise with a fixed binary mask. The mask density was optimized for each network size and for both energy measures separately. This algorithm saved a bit more energy but in large networks the lack of coordination between plasticity input and output layers, again wastes energy Fig 2a, blue curves. For the $M_1$ energy the optimal number of plastic units in the hidden layer is around 50. . .100, irrespective of network size, Fig 2c. In contrast the $M_0$ energy was minimal at a lower number of plastic neurons, preferring fewer, but larger synaptic updates.

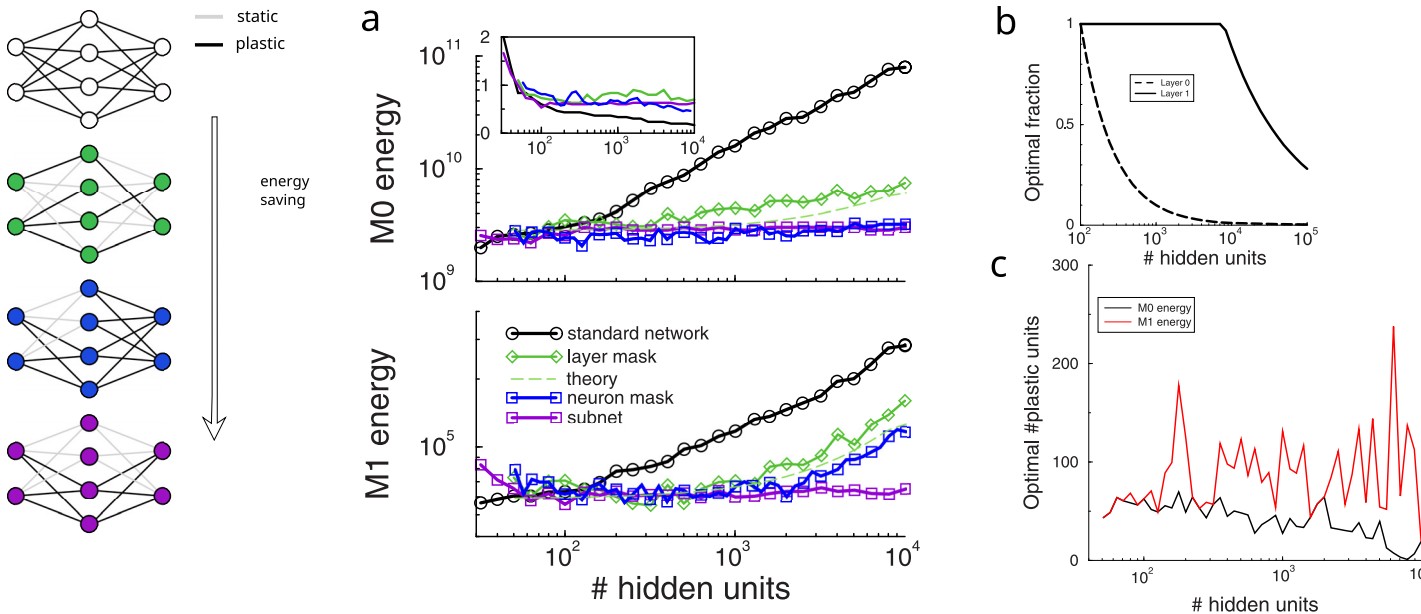

**Fig 2. Restricting plasticity using random masks saves energy.** A: M0 energy (total number of updates) and M1 energy (accumulated changes) for a standard network increases strongly with network size (black, same as Fig 1b). A random synaptic mask saves energy but cannot prevent an eventual increase in large networks (green; theory in dashed green). A bit more energy is saved when only a random subset of neurons in the hidden layer is plastic (blue). Most energy is saved when input and output plasticity is coordinated so that neurons with plastic inputs have plastic outputs, energy becomes independent of network size (purple). B: Theoretically optimal number of plastic synapses (used for green curve in panel a). C: The optimal number of plastic neurons found when using a neuron mask (corresponding to blue curve in a).

There is a trade-off between energy saving and learning speed. In unrestricted networks, training is faster is large networks, but unsurprisingly, with masking training no longer speeds up with larger networks, Fig 2a inset.

## Coordination of plasticity: Subnets

The above algorithm does save a lot of energy compared to standard backprop but very large networks still require more energy. Inspired by the masking algorithm, we coordinate plasticity between input and output of the neuron: if and only if neuron's incoming connections are plastic, then so are its outgoing ones. For large networks this is saves energy because such coordination ensures that energy becomes independent of network size, Fig 2, purple diagram +curves. This has a straightforward explanation: the coordination of incoming and outgoing plasticity effectively constructs a plastic sub-network embedded in a larger fixed network. The energy needed for plasticity is the same as in a network with the size of the subnet, hence the presence of the static neurons do not hinder or facilitate the learning. The optimal size of the plastic subnet weakly depends on the energy measure used (about 60 for $M_0$ and 100 for $M_1$). As can be inferred from Fig 2a, using, say, 1000 neurons for the subnet does not dramatically increase energy.

For networks with many hidden units ($N_h \gtrsim 5000$), it is possible to fix all synapses in the hidden layer, and restrict plasticity to the hidden-to-output synapses, so called extreme learning machines [33, 34]. However, the plasticity in the hidden-to-output synapses still requires energy and we found that when trained with SGD such a setup did not save more energy than the methods presented here. Moreover, it required a precise tuning of the variance of the input-to-hidden weights, which is not required here (not shown).

### Restricting plasticity to synapses with large updates

In an effort to further decrease energy need, we next modified only synapses with the largest magnitude updates [19]. Upon presentation of a training sample, the proposed weight updates in the input-to-hidden layer were calculated via standard back-propagation. However, only synapses with the largest magnitude updates (positive or negative) were competitively selected and modified. Synapses in the hidden-to-output layer were always updated. The set of selected synapses was recalculated for every iteration.

The competition was done in two ways: 1) on the level of each neuron so that only a fraction of synapses per neuron was plastic, 2) across the whole input-to-hidden layer. It is also possible to select only neurons with the largest activities, or neurons with the largest backpropagation error to have plastic synapses. This also saves energy, but not as much as selecting on weight update per layer. The savings are illustrated in Fig 3a for a network with 1000 hidden units. This competitive selection saves $M_0$ energy in particular, Fig 3a.

We wondered if the algorithm in effect works the same as the fixed mask. In other words, does it always select the same synapses to be updated? To examine this we calculated the probability that a given synapses is updated throughout training and extract the inverse Simpson index $q = 1/[N \sum_i p_i^2]$ [35], where $p_i$ is the extracted update probability for synapse $i$. When always the same $k$ synapses would be updated, $q = k/N$. When the updates would be distributed over all synapses, $q = 1$. We find that $q \approx 0.6$ both at the start and end of training, which is much larger than $k/N \approx 0.01$. In contrast to the fixed masks used above, plasticity keeps switching between synapses under this algorithm.

Fig 3b examines the savings across network size, showing large $M_0$ but little $M_1$ savings. The reason for this can be seen in the diagram Fig 3c. The $M_1$ energy, i.e. L1 path length, is independent of the weight trajectory when it does not backtrack (top row). Only when the weight paths backtrack can $M_1$ be saved (middle row), but even this is not guaranteed (bottom row).

Next, we further sought to decrease energy need by selecting only synapses with the largest updates in a subnet. This further decreased the energy needs and again made them independent of network size, Fig 3 (purple curve).

In summary, the most energy efficient plasticity occurs when 1) on a given sample only the largest updates are implemented, and 2) these synapses form a plastic pathway in the network. The competitive selecting via the maximum has another advantage over a fixed mask. While the fixed mask limits the computational power of a network, the maximum selection does not. To show this we used a small network (100 hidden units) and trained for 30 epochs, at which time accuracy had saturated. When using a fixed mask, the restricted network's performance drops with mask size, Fig 3d. However, using either competitive selection mechanism, the maximum accuracy barely dropped.

We wondered if the competition led to fragile performance. Hereto we examined the relation between training error vs testing error, Fig B in S1 Text. If the network generalizes poorly, one would expect low training error but high testing error. Training was done over 30 epochs (so not stopping at 95%). With the competitive algorithm, testing error at a given training error is a bit higher than for vanilla backprop, however, the differences are quite minimal.

**Combining mask with synaptic caching.**   It might appear that we have exhausted all ways to save plasticity energy, but additional savings can be obtained by reducing the inefficiency across trials. During learning the synaptic weights follow a random walk, often partly undoing the update from the previous trial. This is inefficient.

An additional, different way to save plasticity energy exploits that not all forms of plasticity are equally costly. Transient changes in synaptic strength (early phase LTP in mammals, ARM

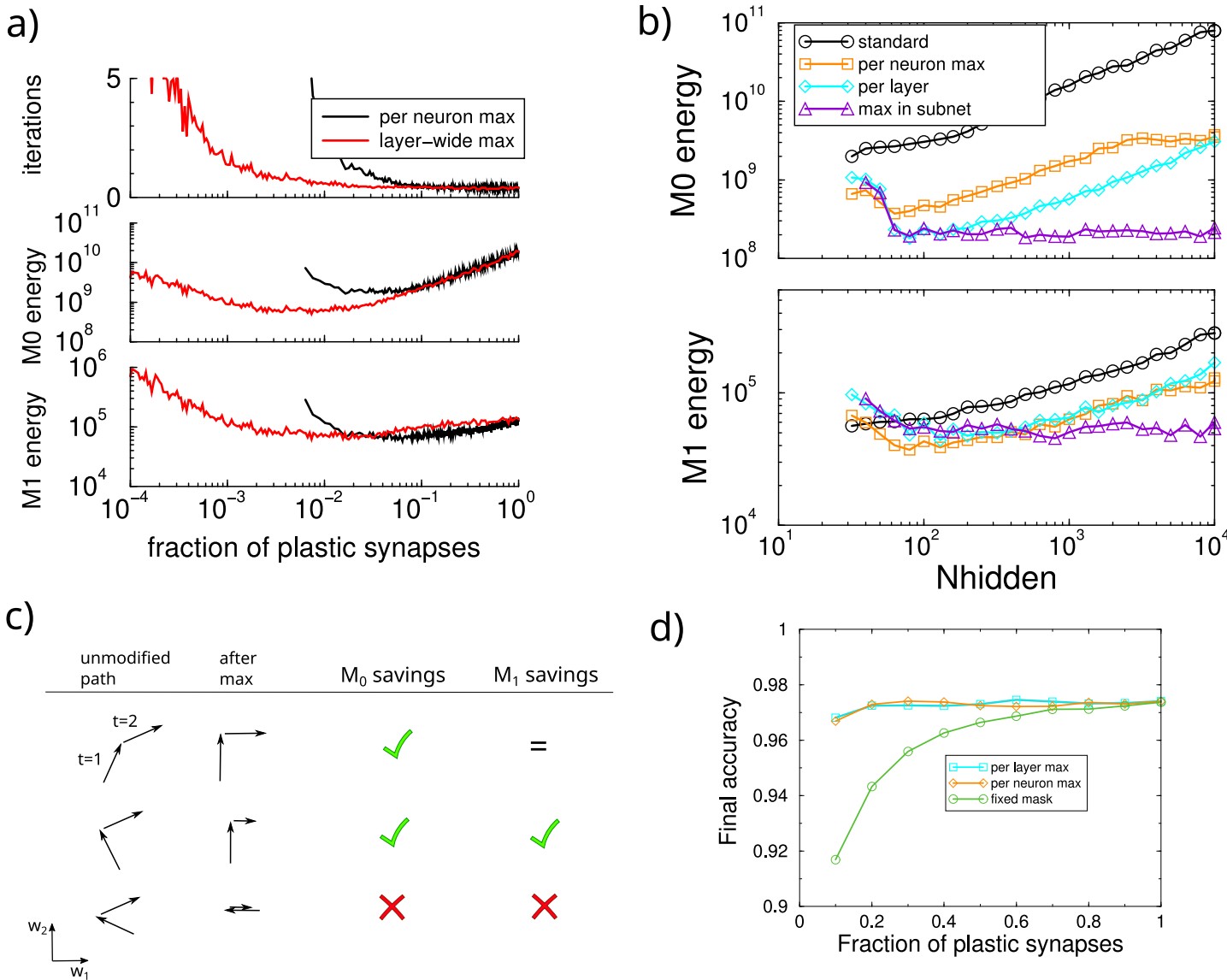

**Fig 3. Energy required for network training when only synapses with large updates are modified.** A: Iterations and energy in a network with 1000 hidden units as a function of the fraction of plastic synapses. The largest updates were selected across the layer (red) or for each hidden layer unit (black). B: Energy requirements as a function of the network size. Allowing only the largest synaptic updates per layer saves substantial amounts of energy (cyan curve), as did only allowing the largest updates per neuron (orange curve), compared to a standard network where all synapses are updated (black curve). The fraction of updated synapses was optimized for each network size. Further savings can be achieved by restricting the selection to subnets (purple), leading to a size independent energy need. C: Diagram of two subsequent weight updates in three hypothetical cases (top to bottom). The maximum selection aligns the updates along the cardinal axes. When the weight trajectory does not return on itself (top), the selection saves $M_0$ energy but $M_1$ energy is identical. In cases where the weight trajectory almost turns back upon itself, there might be no saving (bottom). D: Test accuracy of a small network (100 hidden units) trained to saturation. A fixed mask limits performance, but competitive updating does not.

in fruit-flies) are metabolically cheap, while in contrast permanent changes (late-phase LTP in mammals, LTM in fruit-flies) are costly [6, 7, 9]. It is possible to save metabolic cost by storing updates initially in transient forms of plasticity and only occasionally consolidate the accumulated changes using persistent plasticity. We have termed this saving algorithm *synaptic caching* [36]. It is somewhat similar to batching algorithms in machine learning.

By distributing plasticity over transient and persistent forms, synaptic caching can save energy. The amount of savings depends on how rapid transient plasticity decays relatively to

the sample presentations, and its possible metabolic costs. The largest saving can be achieved when transient plasticity comes at no cost and decays so slowly that all consolidation can be postponed to a single consolidation step at the end of learning.

To explore whether the algorithms introduced in this study are compatible with synaptic caching we implemented both transient and persistent forms of plasticity in the network coded in $w^{\text{trans}}$ and $w^{\text{pers}}$. The total connection strength between neurons was $w^{\text{trans}} + w^{\text{pers}}$. Updates of the weights were first stored in the transient component. Only when the absolute value of the transient component reached a threshold value, the transient weight was added to the persistent weight and the transient component was reset to zero. The decay rate of the transient weights was $10^{-3}$ per sample presentation; the cost of transient plasticity was set to $M_1^{\text{trans}} = c\sum_i |w_i^{\text{trans}}|$, with constant $c = 0.01$ (see [36] for motivation and further parameter exploration).

Fig 4 shows the energy measures as a function of the consolidation threshold. Synaptic caching saves a substantial amount of energy. As the transient plasticity does not contribute to the $M_0$ energy measure, it now just counts the number of synaptic consolidation events. It is lowest at a high consolidation threshold, at even higher thresholds ($\gtrsim 0.06$) the learning no longer converges. By thus limiting and postponing consolidation the $M_0$ energy with synaptic caching is virtually independent of network size.

However, the $M_1$ energy still increases with network size, with a similar dependence on network size as the standard network, Fig 4b. It will still lead to high costs in large networks. Coordinating both transient and persistent plasticity, that is restricting plasticity to subnets, again eliminates this increase. We also tried a variant in which transient plasticity was distributed over the whole network and only consolidation was coordinated, however this did not save energy.

In sum, combining synaptic caching with the above savings strategy saves the most energy. For the $M_0$ energy, synaptic caching with optimal threshold even completely obviates the need

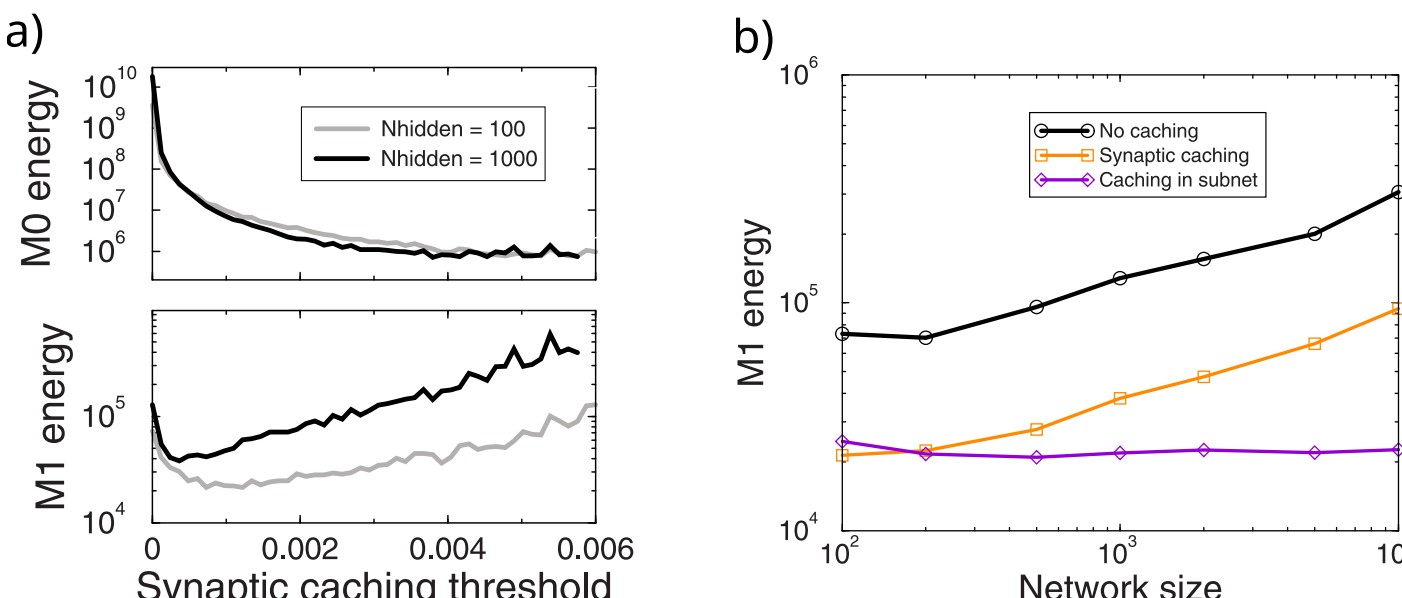

**Fig 4. Synaptic caching.** a) Energy savings achieved by synaptic caching for networks with 100 and 1000 hidden units. The $M_0$ energy that only counts consolidation events, is minimal when the threshold is large, and is independent of network size. The $M_1$ energy trades off cost transient plasticity with consolidation costs and learning time. b) Synaptic caching saves $M_1$ energy. The $M_1$ energy is made independent of size by restricting the plasticity to subnets.

for additional saving strategies in large networks. We emphasize however that synaptic caching does rely on additional synaptic complexity, namely two synaptic plasticity components and a consolidation mechanism. Moreover, if the updates of the transient component would also incur $M_0$ energy, synaptic caching would leave room for additional saving strategies.

## Discussion

Experiments have shown that already simple associative learning is metabolically costly. For neural networks that learn more complicated tasks, energy costs can according to our model become very high, in particular when networks are large. For instance, macaque V1 has some 150 million neurons and some $300 \times 10^9$ synapses [37]. Extrapolating Fig 1, if plasticity were distributed over all these synapses (as backprop prescribes), the number of synaptic updates would be some $10^5$ times larger than required. The $M_1$ energy, which increases less steeply with network size, would still be some 700 times larger, Fig 5. Thus large networks are powerful, but without restrictions the metabolic cost of plasticity could become very large.

We have introduced two approaches to reduce costs. First, restrict plasticity to a subset of synapses. This is most efficient when the plasticity on input and output side of a neuron are coordinated, so that when inputs of a neuron are modified then so are its outputs. Such effects have indeed been observed in experiments, although the precise rules appear complex [38–40].

Second, express plasticity only in synapses with large updates. This is also consistent with neuroscience data, where typically a threshold for plasticity induction has been observed. Our study suggests the presence of a competitive algorithm in which only a certain fraction of synapses is modified. Biophysically, the competition on the neuron level could naturally follow from resource constraints.

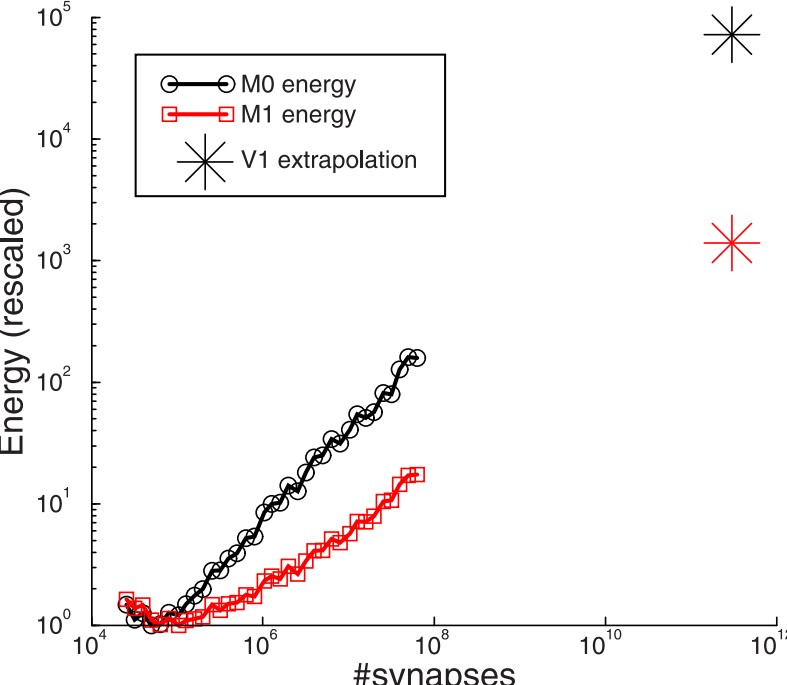

**Fig 5. Extrapolation of results of Fig 1b to the number of synapses in macaque V1.** Naive, unrestricted backprop learning would use $10^5 \times$ more synaptic updates and $10^3 \times$ more $M_1$ energy than minimally required.

These strategies can be combined with our earlier work on synaptic caching. Finally, it also possible to skip over uninformative samples during learning, leading to even further saving [41, 42].

Future studies could include how the saving algorithms should be automatically adapted dependent on task difficulty or network architecture. Another interesting avenue is to find the most energy efficient synaptic modification during transfer learning, c.f. [43].

Given the lack of biological data and the uncertain nature of the main energy consumer, our proposed energy model is currently coarse. In particular the independence assumption is unlikely to be fully correct; yet even the sign of the interactions (cooperative vs competitive) is unknown. We further assumed that the energy is extensive in the number of synapses. It might be that there is a large component independent of synapse number, for example energy needed for replay processes. Such terms, either constant or proportional to the number of training cycles are easily added to the cost model. In that case, the question is at what number of synapses the energy becomes approximately extensive in synapse number. As more experimental data becomes available, the energy model can be refined and the efficiency of the proposed algorithms can be re-examined. Finally, we did not consider the energy required for neural signaling, such as synaptic transmission and spike generation. Ultimately, learning rules should aim to also reduce those costs.

## Supporting information

**S1 Text.** Single file with 2 supplementary figures: Supplementary figure 1: Influence of various assumptions on energy saving. Supplementary figure 2: Relation between train and test error comparing plain backprop to competitive updating.
(PDF)

## Acknowledgments

It is a pleasure to thank Mikhail Belkin, Jiamu Jiang, Simon Laughlin, Thomas Oertner, Joao Sacramento, Long Tran-Thanh, and Silviu Ungureanu for discussion.

## Author Contributions

**Conceptualization:** Mark C. W. van Rossum.

**Formal analysis:** Mark C. W. van Rossum.

**Software:** Mark C. W. van Rossum, Aaron Pache.

**Writing – original draft:** Mark C. W. van Rossum.

**Writing – review & editing:** Mark C. W. van Rossum, Aaron Pache.

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
