## [Decision Letter · Decision Letter 0]

28 Aug 2024

Dear Dr. van Rossum,

Thank you very much for submitting your manuscript "Competitive plasticity to reduce the energetic costs of learning" for consideration at PLOS Computational Biology. As with all papers reviewed by the journal, your manuscript was reviewed by members of the editorial board and by several independent reviewers. The reviewers appreciated the attention to an important topic. Based on the reviews, we are likely to accept this manuscript for publication, providing that you modify the manuscript according to the review recommendations.

In particular, the reviewers have highlighted a few instances where additional description would be useful to clarify some of the details for a less technical audience, or to provide more information about the impact of specific modelling decisions on the results.

Sincerely,

Daniel Bush

Academic Editor

PLOS Computational Biology

Andrea E. Martin

Section Editor

PLOS Computational Biology

Reviewer's Responses to Questions

**Comments to the Authors:**

Reviewer #1: The present paper provides simple and elegant analyses aiming at quantifying the energetic costs associated with standard neural network learning rules, and how biologically plausible methods can reduce them. These are important research questions, with relevance not only to neuroscience, but also to machine learning and neuromorphic engineering. Given that the study considers standard feedforward neural networks trained with stochastic gradient descent methods, its findings will likely appeal to a broad audience.

A possible issue with the papers is whether its sole focus on backpropagation is a problem. While computing gradients by exact backpropagation is biologically questionable, the focus on this scalable, principled gradient-based learning is in my view a strength of the paper, which in fact increases its chances of standing the test of time. There is at present still too much doubt over phenomenological/low-level theories of synaptic plasticity, and how these might enable learning. In line with this, I liked the focus on parsimonious but well argued for energetic cost functions, and on admittedly straightforward, but simple, energy savings methods.

I found the paper clear and a very enjoyable read. I am thus happy to recommend this paper for acceptance, leaving only a few minor questions and suggestions for the author below.

- Test accuracy and training error are usually strongly correlated on MNIST, but this is not always true. More as a sanity check, it would be good to perform some analysis of the training loss, and also to determine whether the energy saving methods converge to a minimizer (in particular for the high capacity, wide networks). Some methods achieve high test accuracy on MNIST, but cannot optimize the network well. In my experience, this is usually a sign of (big) problems, as these methods usually then falter on more difficult tasks.

- Do the results depend on how the network is initialized? There's a brief comment on input-to-hidden variance tuning for fixed hidden layer networks, but it might be worth looking at this in more detail. Note that network initialization can have a big impact on the learning dynamics (cf. "lazy training regime" and NTK theory).

- In numerical optimization, the efficiency of an algorithm is typically analyzed in terms of its rate and order of convergence. It would be nice to discuss the intimate connection between the energy efficiency of learning as defined here and such concepts.

Reviewer #2: Competitive plasticity to reduce the energetic costs of learning

Huge advances are being made in the computations and learning tasks that can be performed by artificial neural networks, but there has been comparatively little focus on the important issue of reducing the energy used by these networks. Performing complex tasks with minimal energy use is something at which the brain excels, and this paper takes inspiration from biological circuits to impose parameters on synaptic modifications within artificial networks, finding two possible ways to drastically reduce the energetic cost of learning.

First the author sets out the energy needed to train standard large network models on the MNIST dataset, with variations in number of hidden units, iterations and learning rate. Two types of energy are defined, relating to total and cumulative synaptic updates. These are then examined while different constraints to synaptic updates are imposed, from random restriction (some improvement in energy use cf standard backprop, but fails for larger networks), to strategic restrictions based on input-output coordination and selecting only those synapses with large weight updates. When the latter restrictions are applied in a competitive rather than fixed manner, the increased energy efficiency does not wane with larger network / mask size.

The author takes a final piece of inspiration from the different timescales of plasticity present in biology. Here, initial learning leads to transient updates whereas consolidation into long-lasting updates is treated separately, implemented in an algorithm they previously coined, “synaptic caching”. This can eradicate some of the inefficient over-writing of updates from previous learning trials, and can lead to even greater energy savings when combined with the strategies above.

The ideas in this paper have widespread importance, their testing is done rigorously and carefully, with results clearly explained and supporting conclusions. I fully support the publication of this article, with only one minor revision:

In the discussion, it is stated that the energy required for synaptic transmission is not considered, which could come across as confusing to a reader with a mostly biological background, as the M0 and M1 energy are specifically related to synaptic updates. I would suggest a “bridging” paragraph in the discussion or intro, which specifically spells out the way that energy use is being considered with respect to how energy is used at biological synapses. The update itself is what is costly - how akin is this to the ATP costs of filament assembly for a new spine, for example? Is there a cost associated with a decreased weight update? These kinds of specifics would make the paper even more widely accessible.

**Have the authors made all data and (if applicable) computational code underlying the findings in their manuscript fully available?**

Reviewer #1: Yes

Reviewer #2: Yes

PLOS authors have the option to publish the peer review history of their article (what does this mean?). If published, this will include your full peer review and any attached files.

Reviewer #1: No

Reviewer #2: No

Figure Files:

Data Requirements:

Reproducibility:

References:

---

## [Decision Letter · Decision Letter 1]

11 Oct 2024

Dear Dr. van Rossum,

We are pleased to inform you that your manuscript 'Competitive plasticity to reduce the energetic costs of learning' has been provisionally accepted for publication in PLOS Computational Biology.

Best regards,

Daniel Bush

Academic Editor

PLOS Computational Biology

Andrea E. Martin

Section Editor

PLOS Computational Biology

Reviewer's Responses to Questions

**Comments to the Authors:**

Reviewer #1: Thank you for replying to my comments.

Reviewer #2: I would recommend this for publication.

**Have the authors made all data and (if applicable) computational code underlying the findings in their manuscript fully available?**

Reviewer #1: None

Reviewer #2: Yes

PLOS authors have the option to publish the peer review history of their article (what does this mean?). If published, this will include your full peer review and any attached files.

Reviewer #1: No

Reviewer #2: No

---

## [Editor Report · Acceptance letter]

23 Oct 2024

PCOMPBIOL-D-24-01138R1 

Competitive plasticity to reduce the energetic costs of learning

Dear Dr van Rossum,

I am pleased to inform you that your manuscript has been formally accepted for publication in PLOS Computational Biology. Your manuscript is now with our production department and you will be notified of the publication date in due course.

With kind regards,

Dorothy Lannert
